# Knowledge about HPV Infection and the HPV Vaccine among Parents in Southeastern Serbia

**DOI:** 10.3390/medicina58121697

**Published:** 2022-11-22

**Authors:** Natasa K. Rancic, Predrag M. Miljkovic, Zorana M. Deljanin, Emilija M. Marinkov-Zivkovic, Bojana N. Stamenkovic, Mila R. Bojanovic, Marko M. Jovanovic, Dusan P. Miljkovic, Sandra M. Stankovic, Suzana A. Otasevic

**Affiliations:** 1Faculty of Medicine, University of Nis, 18000 Nis, Serbia; 2Public Health Institute, 18000 Nis, Serbia; 3Healthcare Center, 18000 Nis, Serbia; 4Clinical Center, 18000 Nis, Serbia

**Keywords:** parental knowledge, HPV infection, HPV vaccine, children, adolescents

## Abstract

*Background and Objectives*: The vaccine against human papilloma virus (HPV) infection is recommended, according to the Serbian National Immunization Program, for children and adolescents aged 9–19 years. Three doses are given keeping in mind the recommendation that the second dose should be administered at least one month after the first dose, and the third at least three months after the second dose. No children who participated in this first study received the third dose because they did not meet these criteria. The study explored parents’ knowledge about HPV infection and their awareness of the HPV vaccine. *Materials and Methods*: A cross-sectional questionnaire-based study was carried out in the city of Nis, in southeastern Serbia. According to the 2011 population census, the sample of children aged 9 to 19 was 850, and during the observed period, 631 children received the vaccine. A total of 615 fully completed questionnaires filled out by parents were included in the study. The study was carried out from 6 June 2022 to 7 October 2022. Multivariable logistic regression analysis was used. The odds ratio (OR) and 95% confidence intervals (CI) were calculated. The statistical significance was *p* < 0.05. *Results*: A total of 615 children were included in the study (499 were vaccinated with the first dose and 116 with the second). Out of 499 children vaccinated with the first dose, 398 (79.6%) were girls, which is significantly higher than the rate for boys (101). The independent variable sex was statistically significant at the level of *p* = 0.84, OR = 2.664 (95% CI from 0.879 to 7.954). Boys are 164% less likely to be vaccinated with the HPV vaccine than girls. We determined that the independent variable place of residence was significant at the level of *p* = 0.041, (OR = 3.809, 95% CI from 1.702 to 8.525). Based on these findings, we determined that parents who came from rural areas were 82% less likely to know about HPV infection and HPV vaccination. Children under 15 years of age were significantly more vaccinated than those ≥15 years (OR = 3.698, 95% CI from 1.354 to 12.598). The independent variable parental education was significant at the level of OR = 0.494, 95% CI from 0.301 to 0.791. Parents who had medical education showed significantly higher awareness about the infection caused by HPV and about the HPV vaccine (*p* = 0.004) than parents with no medical education. The possibility that a parent would decide to vaccinate a child significantly increased upon a pediatrician’s recommendation, *p* = 0.000 with OR = 0.250 (95% CI from 0.127 to 0.707). Health insurance coverage of HPV vaccination for children aged 9–19 years significantly increased the probability of a positive parental decision to vaccinate a child, *p* = 0.001 with OR = 3.034 (95% CI from 1.063 to 8.662). *Conclusion*: We identified several significant factors that were important for HPV vaccination such as: children under 15 years, female sex, urban place of residence, medical education of parents, pediatrician’s recommendation of the HPV vaccination, and HPV vaccination free of charge. Health education and the promotion of HPV vaccination as well as healthy sexual behavior are important factors in the preservation and improvement of the health of the whole population.

## 1. Background

The most common viral infection, which is predominantly sexually transmitted, is caused by human papillomavirus (HPV) [1]. This infection is highly associated with cervical cancer in low- and middle-income regions of the world and mostly with the oropharyngeal cancers in developed regions of the world [1,2]. Cervical cancer is one of the most common malignant tumors among women. It has the fourth highest incidence and mortality rates among women worldwide, with 604,000 new cases and 342,000 deaths in 2020 [1]. The highest prevalence of HPV infection is registered in sub-Saharan Africa (24.0%), Latin America and the Caribbean (16.1%), Eastern Europe (14.2%), and Southeast Asia (14.0%) [3]. In European countries, the prevalence of HPV infection ranges from 2.0% in Spain to 12.0% in Belgium and France [3,4]. About 6.2 million of new HPV infections in females aged 14–44 years are registered in the United States of America (USA) each year [5]. The highest prevalence by age is among nonvaccinated sexually active adolescents [6,7] and among young women under 25 years of age [8,9,10].

Chronic HPV infection, together with other risk factors for cervical cancer, plays an important role in the development of this cancer as well. The most significant risk factors are smoking, immunodeficiency, long-term use of oral contraceptives, promiscuous behavior, and having more than two pregnancies. Other risk factors include a family history of cervical cancer and other descriptive characteristics, obesity, and poor diet [6,7,8,9,10,11]. Cervical cancer is still the second most common malignant tumor in females from 15 to 44 years of age [11]. In most countries, the highest incidence and mortality rates of cervical cancer are registered among younger women under 45 years of age [12], which indicates that this cancer is age-related and mostly affects women in the fertile period. Cervical cancer has a slow progression [13], and if it is discovered at an early stage, it is a highly preventable and highly treatable malignant tumor [14]. It takes more than 10 years after the first exposure to HPV infection to develop some malignant changes in the cervical mucosa [15,16].

There are more than 207 HPV genotypes [16] among which 20 are oncogenic and high-risk HPV types that show high associations with HPV-related cancers, especially with invasive cervical cancer [17]. According to the findings of the first population study of HPV prevalence in women by Kovacevic et al., (2019), the most frequent HPV types in Serbia are type 16 and type 31 [18].

An HPV vaccine was established in order to enable the primary prevention of HPV infection and cervical cancer. Three HPV vaccines exist today: the bivalent (2vHPV) vaccine for girls licensed in 2006, the quadrivalent (4vHPV) HPV vaccine for boys licensed in 2009, and the 9-valent HPV vaccine (9vHPV) licensed for both girls and boys in 2014. The most common World Health Organization (WHO) recommendations regarding routine vaccination are related to children age 9–14 because the HPV vaccine is the most effective if it received before recipients become sexually active, i.e., before they are exposed to HPV infection [19]. The vaccine is also recommended for adolescents, primarily girls and younger women from age 15 to 25 [20]. The CDC recommendations regarding HPV vaccination are somewhat different. Namely, the vaccine is recommended for children aged 11 or 12, but it can also start at the age of nine and is recommended up to the age of 26 [21]. 

In 2007, Australia became one of the first countries to introduce the HPV vaccine [22,23], and in 2020, the HPV vaccine was introduced in more than half of the WHO member countries. For example, the HPV vaccine has been introduced in 85% of the countries in North America and 77% of the countries in Europe [23]. 

The three HPV vaccines do not provide complete protection against all oncogenic HPV types, and immunity is not lifelong [24]. That is why cervical cancer screening is an important preventive measure regardless of vaccine status. There is the well-known Pap smear screening method, which is widely available and inexpensive and has good specificity for the detection of precancerous lesions with a proven impact in reducing cervical cancer prevalence and mortality rates [25]. Additionally, the viral DNA of the high-risk strains of HPV viruses can be easily detected in exfoliated cervical cells using commercially available tests. The HP-DNA test represents a convenient, highly sensitive screening tool and is showing a pattern of becoming the main screening method. However, it has an important flaw reflected in its lower specificity [25].

Organized cervical cancer screening has been conducted in the Republic of Serbia since 2012. The screening is mandatory for women aged 25 to 69 [25], and the HPV vaccine is available. However, cervical cancer remains one of the most prevalent cancers among women [26]. According to the data from the population cancer registry of Central Serbia, cervical cancer was the fourth most common cancer in women in 2020, with 1087 new cases and 453 deaths registered [27]. The age-standardized cervical cancer mortality rates in Serbia are still the highest of all the Balkan countries [4].

In the Republic of Serbia, vaccination against HPV infection is not part of the mandatory national immunization program. It is recommended for children aged 9, before the first sexual intercourse, and primarily for children in the seventh grade of primary school (age 13) [28]. HPV vaccination of children was introduced in Serbia according to the WHO recommendations. Children aged 15–19 may also receive the HPV vaccine free of charge. Until now, organized vaccination against HPV infection has not been conducted among children, adolescents, and young adults in Serbia. This year, the focus is on the vaccination of children and adolescents from the age group 9–19 years. Some factors that impact the coverage of HPV vaccination have been determined in the countries that have been conducting the vaccination for over a decade. These include insufficient population awareness about the HPV infection and its consequences, insufficient recommendations from pediatricians or unclear emphasis on the importance of vaccination, and insufficient parental knowledge. The level of knowledge that parents and adolescents have about HPV infection and vaccination was the subject of a large number of studies in the countries of the European Union (EU). In all countries where the immunization process has started, health policy makers have emphasized that raising the awareness of the population, especially health workers and parents, about the importance of preventing these diseases is crucial in order to better implement vaccination [6,13,17,24]. Having in mind that Serbia is not a member of the EU and that this year is the first time the HPV vaccine is being administered on a more organized and mass basis, the objective of this study was to explore parents’ knowledge about HPV infection and the HPV vaccine. These are the first data about HPV vaccination of children and adolescents 9 through 19 years of age in the city of Nis.

## 2. Material and Methods

### 2.1. Study Design and Study Population

We conducted a cross-sectional study involving the parents of children aged 9–19 years. The period of observation lasted from 6 June 2022 to 7 October 2022. The study took place at the primary clinic at the Healthcare Center Nis at the Counseling Center for Healthy School Children. The Healthcare Center Nis is the largest health care institution of its kind in the Republic of Serbia. The city of Nis is the largest city in southeastern Serbia. An anonymous questionnaire was distributed only to parents of children who came for vaccination. If the parent was not present, which was often the case with children aged ≥ 15, the child did not fill out the questionnaire. Some of the children this age were accompanied by parents because the HPV vaccine is being widely administered for the first time this year, and the children only signed the consent form, which they are legally allowed to do [28].

#### Activities of Pediatricians and Epidemiologists before the Start of Vaccination

Since the beginning of 2022, the promotion of vaccination against HPV infection among parents has been carried out at the counseling center. Certain pediatricians from the Nis health center and an immunization coordinator/epidemiologist from the Nis Institute for Public Health went to Belgrade (the capital of Serbia), where they received basic instructions about the vaccine and were also trained on how to encourage parents and other pediatricians. The results of good preparation were reflected in a great number of parents who brought their children for vaccination in June as well as in unreserved support from all pediatricians for giving this vaccine to as many children as possible. 

The state helped by introducing significant relief by making vaccination free for children aged 9–19 years. According to the 2011 population census, the sample of children aged 9 to 19 was 850. In the observed period, 631 children received the vaccine. This study included 615 fully completed questionnaires, which were filled out by parents, and incomplete questionnaires were excluded.

Before vaccine administration, a visit to the pediatrician at the Counseling Center for Healthy School Children was mandatory as well as signing the consent form. According to expert methodological instructions for the application of Gardasil 9 vaccine in Serbia, consent for children up to 15 years of age had to be signed by parents or guardians. Children over 15 years of age could sign their own consent for vaccination with Gardasil 9 [28]. 

Inclusion criteria were: children born between 2003 and 2013 and signed consent for the vaccination. Exclusion criteria were younger or older children and no signed consent.

Approval by the Ethics Committee of the Public Healthcare Center Nis was not necessary because according to the Serbian National Immunization Laws, signature or written consent of the parent or of the adolescent (if they are aged 15 years or above) is mandatory for the application of each recommended vaccine [28]. 

The questionnaires were administered to parents at the Counseling Center for Healthy School Children. Participation in the study was voluntary. Signed consent forms were obtained from all participants. The parents filled out the short questionnaire during a visit to the pediatrician for the child’s vaccination.

### 2.2. Data Collection

Data were collected using a short semi-structured questionnaire that consisted of two sections (Appendix A).

Section one: The first section explored participations’ sociodemographic characteristics, including sex, age, education, place of residence, and pediatrician recommendation. It consisted of 5 items (Table 1).

Section two: The second section consisted of 15 items divided into three subsections (knowledge about HPV infection, awareness of HPV vaccine, and HPV vaccination knowledge). Knowledge was evaluated with a composite score estimated using a total of 13 items regarding risk and protective factors, preventive measures, and the outcome of HPV infection. Participants had three possible options regarding the proposed factors and correct answers were coded with two points. As for the rest of the questions, the given options were yes, no, and I do not know and correct answers were given two points. The total number of points represented the participants’ HPV infection knowledge score (KS), with higher scores meaning better knowledge. The maximum number of points was 30. The results of all answers are shown in Table 2. Awareness of the HPV vaccine was determined based on whether or not the participants had heard about the vaccine. Participants who had heard about the vaccine answered six more questions about the vaccine (Table 3).

### 2.3. The Nine-Valent (9vHPVvaccine)-Legislative Regulations in the Republic of Serbia

The 9vHPV vaccination in Serbia is recommended for children aged nine and up, before the first sexual intercourse, and primarily for children in the seventh grade of primary school (13 years of age). Active immunization against HPV infection is carried out with the required number of doses (two or three), which is recommended by the WHO depending on the type of vaccine and age [28]. Three doses are given to children aged 9 to 19 keeping in mind the recommendation that the second dose should be administered at least one month after the first dose, and the third at least 3 months after the second dose. All three doses should be administered within one year. All children who participated in this first study did not receive the third dose because they did not meet these criteria. The 9vHPV (6, 11, 16, 18, 31, 33, 45, 52, 58) HPV vaccine, Gardasil 9 is registered for use in females aged 9 to <46 years and males aged 9 to <27 years [22,23]. It gives effective prevention against some premalignant lesions and cancers of the cervix, vulva, vagina, and anus caused by vaccine HPV types and genital warts (Condyloma acuminata) caused by specific HPV types [20]. 

### 2.4. Statistical Analysis

An Excel database was created for all collected data. Data are presented as the mean and standard deviation (SD) or as frequencies and proportions. Percentages were used to describe the demographic status and the frequency of parents’ responses to the questions in the questionnaire. All analyses were performed using SPSS software version 22.0. Multivariate analysis was used to determine the factors that had an impact on the parents’ decision to vaccinate a child. The *p*-value was set at *p* < 0.05.

## 3. Results

Out of the 615 children, 499 received the first dose of HPV vaccine, and 116 received the second dose. Out of the original 499 children (398 girls and 101 boys), there were significantly more girls, 79.6%, than boys. That is, there were 3.9 times more vaccinated girls than boys on average. 

The total number of children aged 9–14 years who were vaccinated with the first dose of HPV vaccine was 280 (56.1%). In the age group 15–19, there were 219 children (43.9%) who received the first dose of the HPV vaccine (*p* < 0.001). 

Table 1 shows the socio-descriptive characteristics of the research participants.

In the age groups 9–14 and 15–19, 3.1 times more girls were vaccinated than boys. There were significantly more children and parents from urban areas than from rural areas, 90.2% vs. 9.8%, respectively, and there were 1.5 times more parents with medical education than non-medically educated parents (Table 1).

The results presented in Table 2 show the percentages of correct answers given by parents about HPV infection where 78.0% correctly listed all the ways of HPV transmission, 68.0% correctly listed the factors that increase the risk of developing cervical cancer, 56.0% correctly answered who is at higher risk of HPV infection, and 82% of parents correctly answered about prevention of HPV infection. Question 10 had the smallest percentage of correct answers, only 48% (Table 2).

All parents of children who received the first dose of the vaccine knew that the vaccine exists, that it was available in Serbia, and the best age to receive it. The smallest number of parents answered correctly about the number of vaccine doses and the duration of immunity. The question regarding the most common side effects had the fewest correct answers (Table 3).

The results of a multivariate regression analysis are presented in Table 4.

The independent variable sex was statistically significant at the level of *p* = 0.84, OR = 2.664 (95% CI from 0.879 to 7.954). Boys are 164% less likely to be vaccinated with HPV vaccine than girls. 

We determined that the independent variable place of residence was significant at the level of *p* = 0.041, OR = 3.809, (95% CI from 1.702 to 8.525). Based on these findings, we determined that parents who came from rural areas were 82% less likely to know about HPV infection and HPV vaccination. Children under 15 years of age were significantly more vaccinated than those ≥ 15 years, OR = 3.698 (95% CI from 1.354 to 12.598). The independent variable parental education was significant, OR = 0.494 (95% CI from 0.301 to 0.791). Parents who had medical education showed significantly higher awareness about the infection caused by HPV and about the HPV vaccine (*p* = 0.004), OR = 0.494 (95% CI from 0.301 to 0.791) than non-medically educated parents. A pediatrician’s recommendation had a significant effect on the vaccination rate, OR = 0.250 (95% CI from 0.127 to 0.707). Health insurance coverage of HPV vaccination for children aged 9–19 years significantly increased the probability of a positive parental decision to vaccinate a child, *p* = 0.001 with OR = 3.034 (95% CI from 1.063 to 8.662) (Table 4).

## 4. Discussion

This is the first organized immunization of children aged 9–19 with the 9-valent HPV vaccine in Serbia. Our cross-sectional questionnaire-based study was conducted with the aim of exploring parents’ knowledge about HPV infection and their awareness of the HPV vaccine. According to the Serbian National Immunization Program, the HPV vaccine is recommended for children and adolescents aged 9–19 years, and for this age group, it is free of charge. In addition to the unexpectedly high response to vaccination against HPV infection, we identified several significant factors that were important for positive parental decisions to vaccinate children with the HPV vaccine: children under age 15, female sex of children, urban residence, medical education of parents, pediatrician recommendation of the HPV vaccination, and health insurance coverage of HPV vaccination.

In general, parents showed good knowledge about HPV infection and the HPV vaccine, which was expected because the majority of parents had medical education. However, we also observed a lack of parental knowledge about whether HPV infection always leads to clinical manifestation of the disease and about who was at higher risk of HPV infection. In terms of knowledge about the HPV vaccine, the questions with the fewest correct answers were those regarding the number of doses, the duration of immunity, and the most common side effects. The results of our study are in agreement with similar studies conducted in Europe and the rest of the world [6,11,29,30,31,32,33]. 

The most common concerns indicated by parents in our study were related to vaccine safety, side effects of the vaccine, and lifelong protection after vaccination. Our results are in accordance with similar studies. Parents in other studies worried more about post-vaccination sexual promiscuity, moral problems related to sexuality, and conservative and religious views, and there was denial that children are at risk [29,34,35].

Our results showed that there were nearly four times more vaccinated girls than boys. This difference in our study is greater than in the literature, where the difference is mostly twofold. Results of meta-analyses showed that there were two times more vaccinated girls than boys [36,37]. The disparities in uptake of the HPV vaccine by sex of child can be a result of the later approval and recommendation of HPV vaccination for boys than for girls, but despite that, many national immunization programs still do not include vaccination of boys. According to the findings of Radisic et al. [38], HPV vaccine uptake among male adolescents is suboptimal. There is a need to address the predictors of uptake by educating parents about boys’ high susceptibility to infection and the benefits of vaccination to reduce the perceived barriers. Data from similar studies show low knowledge about HPV infections and vaccination in the population of adolescents and their parents in both developing and developed regions of the world [6,7,8,10,28,35,36]. In Greece, the HPV vaccine has been part of the national immunization program since 2008, and it is administered for free to girls from 11 up to 26 years of age [10]. The adolescents had insufficient knowledge about HPV infection and about protection measures, as well as about the association between HPV infection and cervical cancer [35].

In our results, there were nearly 3 times more vaccinated children aged 9–14 than adolescents aged ≥15 years, particularly among boys. Parents of children aged 15 years and over were not very well informed about HPV infection and the HPV vaccine. Our findings are in agreement with the findings of other similar studies [20]. Results mainly from systematic analyses available in the literature show that the number of vaccinated children younger than 15 is greater than children older than 15 [37,38].

There are different findings in the available literature. In a systematic meta-analysis by Holman et al., the authors stated that the age of a child was a common reason for refusing or delaying HPV vaccination, and older girls were more likely to be vaccinated than younger girls. In only two studies did age not predict any intention of parents to vaccinate their children [34,35].

The state of parental knowledge has an essential influence on the immunization of children [31]. Newman et al., (2018) presented the results of a meta-analysis in which vaccination was positively impacted by parents’ medical education, knowledge and awareness about the relationship between cervical cancer and HPV knowledge (r = 0.04 (95% CI from 0.04 to 0.13)), and some socio-descriptive variables such as urban versus rural residence (r = 0.10 (95% CI from 0.06 to 0.14)). Fishman et al., found that mothers with more knowledge about HPV were not willing to vaccinate themselves or their daughters [30]. A study of parental knowledge conducted in Poland in 2022 showed that the only factors that really affect attitudes to vaccination are the knowledge and education of parents [17]. The remaining characteristics of parents do not significantly affect the attitude toward vaccination. 

As vaccination progressed in the city of Nis, more parents heard about it but when vaccination began, some parents with children under 15 years of age were away on vacation. Children over 15 were still going to school (high school vacation started on 24 June 2022) and that is why there was initially a greater response from parents with children over 15.

Compared with our study, the literature contains fewer recorded vaccination differences among girls and boys. The results of studies showed that there were two times more vaccinated girls than boys [31,32]. The disparities in uptake of the HPV vaccine by sex can be a result of the later approval and recommendation of HPV vaccination for boys than for girls, but despite that, many national immunization programs still do not include the vaccination of boys [19,20,21]. Our results suggest that it might be more effective to advocate and educate on the HPV vaccination and to strongly promote it to both parents and children.

In undeveloped regions such as Nigeria and Kenya [36,37] and in rural areas and suburban regions, there is resistance towards the HPV vaccination [30]. Surprisingly, in many studies, parents from rural areas were worried about the sexual behavior of their children after the vaccination and did not vaccinate their sons because they were not aware of the direct benefits of HPV vaccination [29,36,37]. This means that socioeconomic status in association with medical education, knowledge, and higher awareness among parents in urban areas might explain their positive decision on the HPV vaccination of children and adolescents. The results of a study from Greece showed that parents’ education level plays a major role in teenagers’ attitudes and beliefs towards the HPV vaccine [10]. In a study conducted in Poland by Smolarczyk et al., (2022), the significant influencing factors in Poland were lack of recommendations and financing of HPV vaccination by the National Health Fund [17].

The findings of 62 studies showed that a significant predictor of successful HPV vaccination was physician’s recommendation (r = 0.46 (95% CI from 0.34 to 0.56)). It had the highest impact on parents’ positive decision, followed by worry about HPV vaccine safety (r = −0.31 (95% CI from −0.41 to −0.16)), preventive examination of children during one year (r = 0.22 (95% CI from 0.11 to 0.33)), and parents’ health beliefs towards vaccines (r = 0.19 (95% CI from 0.08 to 0.29) [39].

In Serbia, the HPV vaccine is expensive, and the government’s approval at the beginning of this year to cover the HPV vaccination for children aged 9–19 with health insurance was important for the success of the vaccination effort. In our results, both pediatrician recommendation and health insurance coverage of HPV vaccination were independently associated with positive decision of parents towards the vaccine. 

Some studies have shown that a pediatrician recommendation was the most significant predictor of HPV vaccination [8,10,17]. According to meta-analysis, health insurance coverage of HPV vaccination (r = 0.16 (95% CI from 0.04 to 0.29)) or lower out-of-pocket cost (r = −0.15 (95% CI from −0.22 to −0.07)) were significant factors for the vaccination [40].

## 5. Conclusions

We here presented the results of the first organized vaccination of children and adolescents against HPV infection, and we also showed the results of the first research on parental knowledge about HPV infection and HPV vaccine in southeastern Serbia. Significant factors for positive parental decisions about HPV vaccination for their children were children under 15 years, female children, urban residence, parents’ medical education, pediatrician recommendation of vaccination, and free vaccination. The vaccination of boys aged 15 years and over was significantly lower compared with girls from the same age group. In accordance with the results of our study, when it comes to the decision to get the HPV vaccine, personal contact with a pediatrician could be very useful both for parents and children, especially for boys. In order to preserve and improve the health of the whole population as well as of groups of individuals at higher risk, the promotion of HPV vaccination, health education measures, the promotion of healthy sexual behavior, and the support of the government are also important.

### 5.1. Strengths of Our Study

This study presented the first results of the organized immunization of children aged 9–19 years with the recommended HPV vaccine. Our findings indicated what factors are associated with positive parental decisions to pursue HPV vaccination for their children. We noted extremely high interest among parents for the first dose of HPV vaccine, and no side effects were reported. This study was created as a longitudinal study, and all vaccinated children and adolescents will be followed up for a minimum of 10 years after the vaccination.

### 5.2. Limitations of Our Study

We could not include all the parents we wanted in the survey.

## Figures and Tables

**Table 1 medicina-58-01697-t001:** The most important socio-descriptive characteristics of the participants.

Characteristics		*n*	%
Age of children	9–14 years	280	56.1
	15–19 years	219	44.7
Sex of children < 15 years	9–14 M	69	24.6
	9–14 F	211	75.4
Sex of children ≥ 15 years	15–19 M	32	9.6
	15–19 F	187	90.4
Place of residence	Urban area	450	90.2
	Rural area	49	9.8
Education of parents	Medical education	299	59.9
	Non-medical education	205	40.1

**Table 2 medicina-58-01697-t002:** Parents’ knowledge about HPV infection.

Question Number	Question	Percentage %
6	How can you get infected with HPV	76.0%
7	What are the main ways of transmission?	78.0%
8	Who is at higher risk of HPV infection?	56.0%
9	Are the HPV infections and malignant diseases connected?	92.0%
10	Does HPV infection always lead to clinical manifestation of the disease?	48.0%
11	Do you know what is a Pap smear test?	89.0%
12	What factors increase the risk of developing cervical cancer?	68.0%
13	How can HPV infection be prevented?	82.0%

%—percentage of correct answers.

**Table 3 medicina-58-01697-t003:** Parental knowledge and awareness about HPV vaccine.

Knowledge about HPV Vaccine	N	% of Correct Answers
1. Is there a vaccine against HPV infection?	499	100%
2. Is it available in Serbia?	499	100%
3. At what age is it best to administer the vaccine?	499	100%
4. In how many doses is it administered?	390	78.2%
5. What are the most common side effects?	321	64.3%
6. How long does immunity last after vaccination?	380	76.2%

N-number of parents who participated; %—percentage of correct answers.

**Table 4 medicina-58-01697-t004:** The significant factors associated with the parental decision to vaccinate children against HPV infection.

Variable		B	95% CI	*p*	OR
Sex	Boys	0.972	0.879–7.954	0.84	2.664
	Girls				
Place of residence	Urban	1.337	1.702–8.525	0.041	3.809
	Rural				
Child’s age	<15	−0.056	1.354–12.598	0.024	3.698
	≥15				
Parental education	Medical education	−0.705	0.301–0.791	0.004	0.494
	Non-medical				
Pediatrician recommendation	Yes	−1.386	0.127–0.707	0.000	0.250
	No				
Health insurance coverage of HPV vaccination	Yes	1.110	1.063–8.662	0.082	3.034
	No				

B–coefficient of regression; CI–Standard Error; *p*-possibility; OR—Odds Ratio.

## Data Availability

The original data can be obtained by contacting the corresponding author.

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
