# Peer review of "Knowledge about HPV Infection and the HPV Vaccine among Parents in Southeastern Serbia"

_medicina, 2022, doi:10.3390/medicina58121697_

Round 1

Reviewer 1 Report

1. It is not original because it is already known that girls over 15 years of age, living in cities, medical education of parents, recommended by pediatricians to be vaccinated against HPV, and that free HPV vaccination can increase vaccination rates.

2. Therefore, the HPV infection landscape in Serbia should be presented more clearly. For example, the uniqueness of this study can only be revealed when it specifically explains why the age criterion is 15, the difference between urban and rural HPV prevalence, and the implications of parental medical education.

3. In particular, it is also necessary to analyze why the vaccination rate for boys aged 9-14 is higher than for girls.

4. It is judged that it will be meaningful only by increasing the number of study subjects.

Author Response

Very Respected Reviewer,

I am very grateful for Your time and very useful suggestions. I tried to correct answer to Your questions.

  1. Research on the knowledge of parents and adolescents about HPV infection and HPV vaccination was the subject of a large number of studies in the countries of the European Union (EU). Having in mind that Serbia is not a member of the EU and that this year for the first time the HPV vaccine is being administered on a more organized and mass basis, the objective of this study was to explore parental knowledge about HPV vaccination and about the HPV vaccine.
  2. We used the age criterions from our professional-methodological guidance of vaccination against HPV infection with Gardasil 9. This guidance used the WHO recommendations for HPV vaccination. The only study of HPV prevalence was done in our North Autonomous Region Vojvodina. We do not know the prevalence of HPV in the whole Serbia. According to the results of the Investigation about health, national Survey in 2019 in Serbia, 2.9% of young people entered into sexual relations before the age of 15. Young persons aged from 15 tо 19 in a 2018 had sexual relations with a periodical partner. Boys stated that they had used a condom during their last sexual intercourse with a periodical partner (61.0% vs.  48.2%).
  3. This ratio boys vs. girls in the age group from 9 to 14 significantly changed when we add more participants.
  4. You are absolutely right. By increasing the sample, we got completely different results. There were 3.1 more vaccinated girls in the age group from 9 to 14 years.
  5. The professional translator checked the English language.

Associate professor Natasa Rancic

Reviewer 2 Report

Abstract:

Line 15: The ending of the sentence could be re-worded to “…325 participants from 9 to 19 years old.”

Line 18: I don’t think the sentence about statistical significance is necessary.

Based on the “Background and objective section”, I was not expecting to see what was presented in the “Results” section.

Line 23: Urban place of residence was significant for what?

Line 24: Define medically educated parents.

The Results section needs to be re-written, it is very confusing for the reader to disentangle some of the findings.

The objective needs to be more clearly stated so that it relates to the findings directly.

Background:

I feel as though there needs to be some re-organization of the Background section for better flow.

-          Paragraph 1- second sentence on association with cancers should be removed.

-          Paragraph 2 should delineate how many infections there are (currently paragraph 3) – this also needs more information about infections in males or other type of information.

-          Paragraph 3 should be the current paragraph 2 discussing the various HPV genotypes.

-          Paragraph 4 should start with the sentence from Paragraph 1 [lines 39-41 “This infection is highly associated with…” The paragraph can then continue on as is with numbers of diagnoses.

-          Paragraph 5 should be the current paragraph 7.

-          Paragraph 6 can stay as is with discussion about vaccine.

-          The new paragraph 7 should be the current paragraph 8.

Line 61: In what country is cervical cancer the 19th most common cancer?

Line 62: This sentence seems to contradict the prior one.

Line 66: The end of the sentence should read “if it is discovered at an early phase…”

Line 68: Should read “…after the first exposure to the HPV infection...”

Line 71: Please add a period after “…was established.”

Lines 70-76: I think it’s important to note that not all 3 vaccines are in use today and to name which is the only one still on the market.

Lines 88 – 95: It seems like some of this information should have gone in the paragraph that discusses the diagnoses (Paragraph 4).

Again, I think the objective needs to be re-written because as it is currently written, I only expect to see data on knowledge and awareness of the vaccine and nothing else.

 Materials and Methods:

Section 2.1: Did these individuals have some type of health insurance? Also, maybe describe the population of Nis a bit more.

Section 2.3: I think a lot of this could have gone in the Background section.

Section 2.4: Were any descriptive statistics used?

For this section, I didn’t have a clear understanding of what would happen once a person walked into the hospital. Who approached them to be part of the study? Was it anyone who asked for a vaccine during that time period?

Results:

In the first paragraph, make sure to mention the overall vaccination rate before going into detail about other things.

Lines 147-148: It states 42.2% of 9-14 year olds were vaccinated, but in the table it says 42.8%. Also for those 15-19 it says 57.8%, but in the table it says 57.2%. Please correct.

What is presented in results is not accurately reflected in the Objective statement at the end of the Background section – as mentioned before, the objective needs to be re-worked.

Table 1: P-values should be included.

Lines 153-157: Just my opinion, but may be better to just present these as percentages as they are in the table.

Table 2: Age of the child – It is supposed to be <15 years of age and >= 15 (it currently shows <=15)?

Lines 162-177: I do not see the confidence intervals listed in the table, but they are mentioned throughout the paragraph. If you are going to include these in the narrative, is best to have the corresponding numbers in the table.

Discussion:

First sentence should be re-written objective to include vaccination rates and factors associated with vaccination rates.

Line 196: Should read “…with other similar literature [or studies].”

Lines 203-205: In the Holman et al. study, did they find that younger age was a common reason for refusing/delaying or older age?

Lines 210-211: This information was not presented in the Results section so out of place here.

Lines 213-220: This again seems like new information that hadn’t been discussed in the Background section. I’m not sure that it should be introduced here.

Lines 228 – 252: All of these findings need to be better integrated as a difference or similarity to what was found in this current study.

Overall, the Discussion section needs a lot of work to tighten up what the main findings are and how they directly compare to what been found in other studies, primarily with similar populations if possible. There also needs to be a section on next steps or future studies.

Conclusion:

Line 268: Should read “…positive parental decision…”

Line 269: Should read “We noted extremely high interest by parents…”

Line 270: The study design was declared as a cross-sectional study, it should have been noted in Methods that this study was embedded in a longitudinal study.

Lines 274-275: Please elaborate and explain why all parents and children were not included.

References:

A lot of the references were over 10 years old. Should try to find more recent sources.

Author Response

Very Respected Reviewer,

I am very grateful for Your time to read my manuscript. I am grateful for all usefull sugstions.

  1. I re-write the whole Introduction. I removed all contradictory sentences.
  2. I the objective and the rsults. I added more new data upto the 7 October 2022. The sample is bigger and the results of statistical analyses are different.
  3. Now the title, the objective, the findings and the conclusion of the manuscript are in accordance.
  4. Materials and Methods.

Section 2.1: Did these individuals have some type of health insurance?

Yes, they have. In the Republic of Serbia, vaccination against HPV infection is not part of the mandatory national immunization program. In Serbia, the HPV vaccine is expensive and the government’s approval at the beginning of this year to cover the HPV vaccination only for children aged 9-19 by health insurance was important for the success of the vaccination.

  1. Was it anyone who asked for a vaccine during that time period?

The study took place in the primary health institution of the Health Center Nis at the Counseling Center for Healthy School Age Children and each parent who get child to the vaccination and who want to participate (to fill in a short questionnaire) was included in the study. Until now, organized vaccination against HPV infection has not been conducted among children and adolescents in Serbia. This year the focus is on the vaccination of children and adolescents from the age group 9-19 years. All parents were very motivated to know something more about the HPV vaccine and to participate in the study. They also have many questions.

  1. Results Added new data, new numbers, all section re-written
  2. Discussion

Please elaborate and explain why all parents and children were not included.

According to the 2011 population census, the sample of children aged 9 to 19 was 850. In the observed period, 631 children responded to vaccination. A total of 615 questionnaires were fully completed and only those were included in the study.

  1. References:

A lot of the references were over 10 years old. Should try to find more recent sources.

I added new references. There are 70% of all references which are not older than 5 years.

  1. The professional translator checked the English language.

Associate professor Natasa Rancic

Reviewer 3 Report

The manuscript "Prevention of Human Papillomavirus Infections in Children 2 and Adolescents: Vaccination in Focus" is very well written  but suffers few scientific flaws which need to be addressed before a conclusion can be reached. 

1. The questionnaire of the study must be either supplemented or provided in the manuscript. 

2. The study period mentioned is less, I think more data is need to verify the statistics. 

3.  The manuscript needs to recheck for English language. 

Author Response

Very Respected Reviewer,

I am very grateful for Your time and very useful suggestions. I tried to correct answer to Your questions.

  1. The questionnaire of the study must be either supplemented or provided in the manuscript

I added the questionnaire to the manuscript and as Supplementary file.

  1. The study period mentioned is less, I think more data is need to verify the statistics. 

The observed period is longer from 6 June 2022 up to 7 October 2022.

  1. The manuscript needs to recheck for English language. 

 The professional translator checked the English language.

Round 2

Reviewer 1 Report

This study is considered a necessary subject for the national situation of Serbia. Complementing the results by adding a study sample is highly appreciated, and I hope that it will serve as an opportunity to inform the need for HPV vaccination.

Author Response

Very Respected Reviewer,

I added some sentences to the sections Background and Material and Methods.

Background

  1. Chronic HPV infection, together with other risk factors for cervical cancer, plays an important role in the development of this cancer as well. The most significant risk factors are smoking, immunodeficiency, long term use of oral contraceptives, promiscuous behavior, and having more than two pregnancies. Other risk factors include a family history of cervical cancer and other descriptive characteristics, obesity and poor diet [6-11].

  1. An anonymous questionnaire was distributed only to parents of children who came for vaccination. If the parent was not presents, which was often the case with children aged ≥ the child did not fill out the questionnaire. Some of the children this age were accompanied by parents because the HPV vaccine is being widely administered for the first time this year, and the children only signed the consent form, which they are legally allowed to do [28].

The questionnaires were administered to parents at the Counseling Center for Healthy School Children. Participation in the study was voluntary.

  1. Three doses are given to children aged 9 to 19 keeping in mind the recommendation that the second dose should be administered at least one month after the first dose, and the third at least 3 months after the second dose. All three doses should be administered within one year.

…vaccination is recommended up to the age of 26 [21]. The 9vHPV (6, 11, 16, 18, 31, 33, 45, 52, 58) HPV vaccine, Gardasil 9, is indicated for active immunization of both females up to 45 [28].

  1. A professional translator re-corrected the English language in the paper.
  2. Tables have been redone, references have been sorted.

Reviewer 2 Report

I noticed the term “parenteral knowledge” is used throughout as a change; however, I’m wondering if the authors mean “parental knowledge.”

Abstract:

Lines 19-20: Seems like this is repetitive.

For those children over 15 years old, they should receive a third dose, however I did not see that mentioned in the abstract.

Background:

I had previously suggested that paragraph 3 should discuss the connection between HPV and cervical cancer. As it stands now, the transition is abrupt and for readers not aware of the connection, it is not apparent.

Lines 96-110: As far as I know, only Gardasil 9 is available – you may want to re-word this paragraph as it is written as though all 3 vaccines still exist.

Line 109: CDC guidelines state that both men AND women can receive the vaccine up to 26, and even up to 45 years old with shared decision making with a provider.

Materials and Methods:

Section 2.2: Were children over 15 years old able to complete the survey or their parents had to? It state that children over 15 could sign out the consent, but stated later that parents completed the survey. Please clarify.

Results:

Line 237: The first sentence is choppy and needs to be expanded – response rate for what? Again why did nobody over 15 years old get the recommended third dose?

Discussion: No comments

Conclusion: No comments

References: No comments

Overall, I would ask that the next revised draft be sent without track changes as, especially with the tables, it made it very difficult to read. Also, this needs to be edited carefully, as I found many typos, misspellings, and grammatical errors.

Author Response

I noticed the term “parenteral knowledge” is used throughout as a change; however, I’m wondering if the authors mean “parental knowledge.”

  1. I correct that to parents‵ knowledge

Abstract:

Lines 19-20: Seems like this is repetitive.

For those children over 15 years old, they should receive a third dose, however I did not see that mentioned in the abstract.

  1. There were no children who should have received the third dose during the observed period. All children who participated in this first study did not receive the third dose because they did not meet these criteria.

Background:

I had previously suggested that paragraph 3 should discuss the connection between HPV and cervical cancer. As it stands now, the transition is abrupt and for readers not aware of the connection, it is not apparent.

  1. Chronic HPV infection, together with other risk factors for cervical cancer, plays an important role in the development of this cancer as well. The most significant risk factors are smoking, immunodeficiency, long term use of oral contraceptives, promiscuous behavior, and having more than two pregnancies. Other risk factors include a family history of cervical cancer and other descriptive characteristics, obesity and poor diet [6-11].

Lines 96-110: As far as I know, only Gardasil 9 is available – you may want to re-word this paragraph as it is written as though all 3 vaccines still exist.

  1. I changed that paragraph. According to the findings of the first population study of HPV prevalence in women by Kovacevic et al. (2019), the most frequent HPV types in Serbia are type 16 and type 31 [18].

Line 109: CDC guidelines state that both men AND women can receive the vaccine up to 26, and even up to 45 years old with shared decision making with a provider. .

…vaccination is recommended up to the age of 26 [21]. The 9vHPV (6, 11, 16, 18, 31, 33, 45, 52, 58) HPV vaccine, Gardasil 9, is indicated for active immunization of both females up to 45 [28].

Materials and Methods:

Section 2.2: Were children over 15 years old able to complete the survey or their parents had to? It state that children over 15 could sign out the consent, but stated later that parents completed the survey. Please clarify.

  1. An anonymous questionnaire was distributed only to parents of children who came for vaccination. If the parent was not presents, which was often the case with children aged ≥ the child did not fill out the questionnaire. Some of the children this age were accompanied by parents because the HPV vaccine is being widely administered for the first time this year, and the children only signed the consent form, which they are legally allowed to do [28].

The questionnaires were administered to parents at the Counseling Center for Healthy School Children. Participation in the study was voluntary.

Results:

Line 237: The first sentence is choppy and needs to be expanded – response rate for what? Again why did nobody over 15 years old get the recommended third dose?

  1. Three doses are given to children aged 9 to 19 keeping in mind the recommendation that the second dose should be administered at least one month after the first dose, and the third at least 3 months after the second dose. All three doses should be administered within one year.

Discussion: No comments

Conclusion: No comments

References: No comments

Overall, I would ask that the next revised draft be sent without track changes as, especially with the tables, it made it very difficult to read. Also, this needs to be edited carefully, as I found many typos, misspellings, and grammatical errors.

  1. A professional translator re-corrected the English language in the paper.
  2. Tables have been redone, references have been sorted.
        1.  
